# Quantitative Trait Loci Mapping for Bacterial Wilt Resistance and Plant Height in Tomatoes

**DOI:** 10.3390/plants13060876

**Published:** 2024-03-19

**Authors:** Muhammad Irfan Siddique, Emily Silverman, Frank Louws, Dilip R. Panthee

**Affiliations:** 1Mountain Horticultural Crops Research and Extension Center, Department of Horticultural Science, North Carolina State University, 455 Research Dr., Mills River, NC 28759, USA; 2Department of Plant Pathology, North Carolina State University, Raleigh, NC 27695, USA

**Keywords:** bacterial wilt, quantitative trait loci, *Solanum lycopersicum*, *Ralstonia solanacearum*, plant height

## Abstract

Bacterial wilt (BW) of tomatoes, caused by *Ralstonia solanacearum*, is a devastating disease that results in large annual yield losses worldwide. Management of BW of tomatoes is difficult due to the soil-borne nature of the pathogen. One of the best ways to mitigate the losses is through breeding for disease resistance. Moreover, plant height (PH) is a crucial element related to plant architecture, which determines nutrient management and mechanical harvesting in tomatoes. An intraspecific F_2_ segregating population (NC 11212) of tomatoes was developed by crossing NC 84173 (tall, BW susceptible) × CLN1466EA (short, BW resistant). We performed quantitative trait loci (QTL) mapping using single nucleotide polymorphic (SNP) markers and the NC 11212 F_2_ segregating population. The QTL analysis for BW resistance revealed a total of three QTLs on chromosomes 1, 2, and 3, explaining phenotypic variation (R^2^) ranging from 3.6% to 14.9%, whereas the QTL analysis for PH also detected three QTLs on chromosomes 1, 8, and 11, explaining R^2^ ranging from 7.1% to 11%. This work thus provides information to improve BW resistance and plant architecture-related traits in tomatoes.

## 1. Introduction

The bacterial wilt (BW) of tomatoes, caused by *Ralstonia solanacearum*, is a devastating disease that results in substantial annual crop losses worldwide. Over 200 plant species are susceptible to BW, including several members of the *Solanaceae* family [1,2,3,4]. *Ralstonia* can survive in soils, crop debris, and weed species under favorable environmental conditions [5]. This economically impdisease attacks plants through roots in the soil. The soil-borne bacterium traverses the root cortical tissue to reach the xylem and reproduce exponentially before the invasion of the water column [6,7,8]. Once *Ralstonia solanacearum* is in the xylem, the bacteria multiply and spread through the water conduct system to the growing points, preventing water movement to the whole plant [9,10]. A recent study proposed four bottlenecks that restrain the bacterial movement in resistant tomatoes: (1) root colonization; (2) vertical movement from roots to shoots; (3) circular vascular bundle invasion; and (4) radial apoplastic spread in the cortex [11]. Symptoms of BW include wilting, yellowing of foliage, adventitious root formation, and vascular browning of infected plants [12].

Management of BW of tomatoes is difficult, and growers frequently rely on integrated pest management tactics such as crop rotation, cultural practices, fumigation, host resistance, and grafting with disease-resistant rootstocks [13,14,15,16,17,18]. Even so, control is frequently inadequate or too expensive. Host resistance is a sustainable management tool for controlling BW; however, there are limited sources of resistance available [19]. Some BW-resistant cultivars have been released, including ‘Neptune’, ‘Saturn’, and ‘Venus’ [20]. Tomato resistance to BW is strongly associated with small fruit size and other undesirable vegetative characteristics [19,21,22,23], limiting the commercial adoption of resistant cultivars worldwide [20]. BW resistance has not proven durable against the diversity of strains present in different geographic regions [21,24,25,26]. Artificial inoculation experiments with multiple *Ralstonia* strains revealed that breeding lines CRA66, HI7996, and HI7998 had varying levels of resistance, and resistance was often strain-specific [21,22,23]. Resistance is strain-specific in the HI7996 breeding population in Taiwan [27].

Tomato resistance to BW is quantitative, relying on several minor resistance genes to protect against bacterial invasion [19]. Accurate phenotyping of quantitative resistance is difficult because disease responses range from 0–100, and precise phenotyping requires screening many biological replications under consistent environmental conditions to observe the range of disease response [2]. BW screening accuracy can also be impacted by plant density, age, and size, as well as environmental factors including temperature and soil moisture [12,28,29]. Under cooler temperature conditions, BW disease incidence may be suppressed, whereas higher air and soil temperatures favor disease development [30]. Maintaining consistent environmental conditions in the greenhouse can be difficult and near impossible under field conditions; however, controlled environmental growth chambers have proven effective in providing reliable disease response [31,32,33,34,35].

Several molecular marker systems have been used for mapping resistant loci against BW and other traits in tomatoes, such as amplified fragment length polymorphic (AFLP), simple sequence repeat (SSR), cleaved amplified polymorphic sequences (CAPS), sequence-characterized amplified region (SCAR), and restriction fragment length polymorphisms (RFLP) [26,34,36,37,38,39,40,41,42,43]. These QTL mapping studies revealed different loci depending on the sources of resistance and pathogen strains. Two resistant QTLs were detected on chromosomes 6 and 10 by [38]. Other studies using F_3_ and F_2:3_ populations indicated that resistance loci were positioned on five different chromosomes (3, 4, 6, 8, and 12) [26,34,37]. Another study reported two main effect QTLs (*Bwr-12* and *Bwr-6*) using an F_9_ recombinant inbred line (RIL) population developed from a cross between HI7996 (resistant) and WVa700 (susceptible). These two major QTLs contributed 56% (*Bwr-12*) and 22% (*Bwr-6)* phenotypic variation [26,34,41,42]. A recent image-based non-destructive phenotyping report identified QTLs on chromosomes 2, 3, 4, 6, 8, 10, and 12 [44]. However, most of those studies used non-US *Ralstonia* strains.

Plant height (PH) is an important agronomic trait in crops. Height can impact the architecture of plants, nutrient and water management, and mechanical harvest ability, which ultimately affects the quality and yield [45]. In fresh-market tomatoes, a compact growth habit is desirable because it enables mechanical harvesting [45,46,47]. PH is a quantitative trait. Several QTLs have been identified related to PH in tomatoes [45]. A study detected nine QTLs (*ht1*, *ht3*, *ht5a*, *ht5b*, *ht6*, *ht7*, *ht9*, *ht10*, and *ht11*) linked to PH using a segregating population developed from a cross between interspecies tomatoes (vendor Tm-a2 and LA716) [48]. A similar study reported a main effect of QTL on chromosome 2 controlling PH in a backcross population derived from *Lycopersicon esculentum* cultivar ‘M82-1-7’ and *L. pimpinellifolium* (LA1589) [49]. Another study reported several QTLs on chromosomes 2, 3, 4, 6, and 7 related to PH using RFLP markers in a RIL population [50]. QTLs have reportedly controlled PH on chromosomes 3, 4, 9, 11, and 12 in 20 introgression lines derived from *Solanum chmielewskii* introgression into *S. lycopersicum* [51]. More recently, three QTLs *h4t2a*, *h4t3a*, and *h4t7a* were reportedly linked to PH in an introgression line population derived from *Solanum pennellii* (LA0716) introgression into an inbred line (1052) [45,52], and seven QTLs related to PH were detected on chromosomes 1, 3, 5, 9, 10, 11, and 12 in an F_2_ population. However, the PH QTLs have not been mapped using SNP-based molecular maps, which can facilitate robust marker development for selection.

The NCSU tomato breeding program has developed several populations for disease resistance to prevalent soil-borne diseases, including BW, and to improve horticultural traits such as PH. However, developing BW-resistant tomato cultivars is a cumbersome task that is confounded by a lack of resistant genetic resources, unstable resistance in diverse geographic areas, and multiple *Ralstonia* strains, as discussed above. There are several ways to generate genetic variation for disease resistance in a breeding population; however, the most common method is by bi-parental (BIP) crossing with contrasting parents (i.e., susceptible and resistant parents) to develop segregating populations for multiple traits that represent a unique genetic recombination. In the present study, experiments were conducted using the SNP marker-based linkage map to identify the QTL associated with BW and PH in intraspecific tomato populations derived from NC 84173 × CLN1466EA.

## 2. Materials and Methods

### 2.1. Population Development and Pedigree

The intraspecific population of tomatoes was developed by crossing NC 84173 × CLN1466EA. The resistant parent, CLN1466EA, is an Asian Vegetable Research and Development Center (AVRDC) line (provided by Dr. Peter Hanson) derived from a double cross of (CL5915-2-6-3-3-0-4 × CRA84-26-1-3) × (CLN399-19-6-13-12-4 × CRA84-26-1-3). This line was developed for BW resistance, TMV resistance, heat tolerance, and large-size fruit for the fresh market tomato industry. CLN5915-2-6-3-3-0-4 is a heat-tolerant line from a CLN5915 cross. CLN399-19-6-13-12-4 is another heat-tolerant line with BW and TMV resistance (*Tm2a* gene). CLN5915 was used in breeding the CLN399 lines. The CRA84-26-1-3 is an inbred line derived from the INRA Guadeloupe breeding program. The CRA84-26-1-3 inbred line has BW resistance and large-size fruit. The susceptible parent, NC 84173, is an inbred line developed by Dr. Randy Gardener at the NCSU Tomato Breeding Program in Mills River, NC. The inbred line NC 84173 was developed from a cross of (NC8276 (X)-12-10C X Fla. 7060–ESBK-1C) as a single plant selection in the F_2_ to F_3_ generations. This line was developed for Verticillium wilt and Fusarium wilt resistance (Race 2), superior fruit quality, and large fruit size.

### 2.2. Growing Conditions

Individual plants (140) were sampled from the F_2_ population (250) for mapping QTLs. The plants were grown in a controlled greenhouse at the NCSU Phytotron. The greenhouse temperature was maintained at 26 °C/22 °C day/night temperatures. The greenhouse light conditions were 14 h/ 10 h for day/night. Seeds were sown into Fafard 2P media in 48-cell trays and placed in a greenhouse. Seedlings were transplanted 28 days post-sowing (DPS) to 2-gallon pots to be phenotyped, self-pollinated, and harvested to produce F_2:3_ families. Plants were watered twice a day with deionized water and received a nutrient solution once a day for three days per week. The nutrient solution was prepared by the Phytotron [53]. Plants were top-dressed with a 200 ppm calcium nitrate solution twice a week once the fruit set occurred. Weekly pH and soluble salt readings ranged from 5.15 to 6.55 pH and 400 EC to 650 EC, respectively.

### 2.3. Plant Height Evaluation

Individuals (140) were evaluated for plant height according to the tomato descriptor developed by the International Plant Genetic Resources Institute [20]. Plant height (the distance from the base of the plant to the top of the main stem) was calculated from the boundary between the soil and the tomato plant to the fourth truss when the fruits on the fourth truss were at the half-maturing stage in a centimeter (cm) scale. 

The visual illustration of the correlation matrix and principal component analysis (PCA) was performed by using the R language v3.2.3 coupled with the RStudio interface v1.0.143 and R packages (“FactoMineR”, “factoextra”, “ggplot2”, “ggplots”, “corrplot”), respectively [54,55].

### 2.4. Bacterial Wilt Resistance Evaluation

Bacterial cultures were prepared 48 h before inoculation. Individual plants were clonally multiplied by cuttings to represent biological replication for disease screening. Jackson County isolate was collected from a field site in Whittier, NC, and used for all inoculation experiments. The isolate was stored in −80 °C cryogenic storage and cultured with a disposable loop depositing 10 μL of bacterial suspension onto Casmino Peptone Dextrose Agar media (CPD). Bacterial cultures were incubated at 30 °C for 48 h in the dark. Colonies exhibiting characteristic milky, fluidal, and irregularly shaped morphology were selected, and the inoculum concentration was adjusted in sterile deionized water to 10–8 cfu/mL after examining OD600 readings with a spectrophotometer. The inoculum was prepared on the day of inoculation and maintained at room temperature (21 °C) until inoculation. The soil drench inoculation method was used to artificially inoculate rooted cuttings. Susceptible and resistant parents (NC 84173 and CLN1466EA) were included in inoculation experiments to validate screening results. Each rooted cutting was wounded in the root zone on two sides of the main stem with a sterile scalpel. Inoculum (10 mL) was pipetted into the wound site of each clone [33]. Ten biological replicates were used for each line. Plants were scored for disease incidence at 6 DPI and then every other day until 21 DPI, when the experiment was terminated. Plants were scored on a scale of 0 to 4 [28,56]. A score of 0 represented no disease, a score of 1 was equivalent to one leaf wilting, a score of 2 represented half of the plant wilting, a score of 3 exhibited whole plant wilt, and a score of 4 indicated the whole plant was severely wilted and dead [56]. The area under the disease progress curve (AUDPC) was calculated as described by [57].

### 2.5. DNA Extraction and Genotypic Analysis

DNA extraction was performed with a DNeasy mini plant kit (50) (Quigen Inc., Venlo, The Netherlands). Young leaf tissue was collected for DNA extractions. The extraction protocol was followed according to the manufacturer’s instructions, Quick-Start Protocol, provided in the Quigen kit. Genotyping was conducted using Illumina^®^ microarray Beadchips (Illumina^®^, San Diego, CA, USA). The F_2_ individuals (140) and parents (CLN1466EA and NC 84173) were genotyped using single nucleotide polymorphism (SNP) markers. The microarray Beadchips contain 7720 SNPs that hybridize and fluoresce when polymorphisms are detected, and the lamination for each SNP marker was recorded. The image generated from microarray SNP genotyping was processed with GenomeStudio software (version 1.0), and then data were generated for each SNP marker position. The data were sorted based on whether they were homozygous or heterozygous, cleaned, and then analyzed. A total of 7720 SNP markers were screened against the 140 individuals, parents, and controls at the Michigan State University Genomic Core Facility.

### 2.6. Linkage Map Construction and QTL Mapping

The linkage map was developed using JoinMap 4.0 [58]. The grouping mode was placed as the independent limit of spotting, and the mapping algorithm was used to perform regression mapping (limit of detection > 2.5, recombination frequency < 0.4, and jump = 5) [59]. The Kosambi mapping function was employed to translate recombination frequencies into map distance. Independent limit of detection and maximum likelihood algorithms were used to group and order markers, respectively. The arrangement of the markers within each chromosome was according to the recombination events between the markers. The QTL analysis was performed through Windows QTL Cartographer v 2.5 software [60]. The Composite Interval Mapping (CIM) process was utilized with the default parameters (Model 6). A backward regression was used to perform the CIM analysis to enter or remove background markers from the model. The walking speed was fixed at one cM for the discovery of QTL. This software also obtained the additive effect and the proportion of the phenotypic variation (R^2^-value) for each QTL. A 1000 permutation option was selected to define the likelihood of an odd (LOD) score threshold to ascertain the presence of QTLs [61,62]. The AUDPC values and PH mean values were used for the QTL analysis.

## 3. Results

### 3.1. Phenotypic Evaluation for Bacterial Wilt Resistance

A total of 140 F_2_ individuals (obtained from the cuttings) with parental lines were evaluated in artificial inoculation experiments at the North Carolina State University Phytotron. Bacterial wilt (BW) incidence and severity varied among F_2_ individuals; however, susceptible genotypes consistently displayed high disease incidence and severity, whereas resistant genotypes consistently displayed low disease incidence (Figure 1A). The AUDPC values varied among the segregating population (Figure 1A). The lowest AUDPC values were observed at 2.5, while the highest was 12.2 in the population (Figure 1A). The susceptible parent NC 84173 displayed a 7.1 AUDPC value, and the resistant parent CLN1466EA had an AUDPC value of 3.2. The distribution of phenotypic data was continuous, indicating quantitative and polygenic control of BW resistance in tomatoes (Figure 1A).

### 3.2. Phenotypic Evaluation for Plant Height

Plant height (PH) was calculated in centimeters (cm) from the boundary between the soil and the tomato plant to the fourth truss when the fruits on the fourth truss were at the half-maturing stage. The PH measurement values showed a variation among the segregating population (Figure 1B). The highest PH was measured at 134 cm, while the shortest measurement was recorded at 20 cm (Figure 1B). The parental line NC 84173 exhibited an average PH of 84.3 cm, whereas the parental line CLN1466EA had an average PH of 67.3 cm (Figure 1B). The phenotypic data showed a continuous distribution, indicating quantitative and polygenic control of PH in tomatoes (Figure 1B).

### 3.3. Correlation Analysis and PCA

The correlation matrix indicated a connection between BW resistance and PH (Figure 2A). The correlation analysis revealed a negative correlation between BW resistance and tomato PH (Figure 2A). The PCA approach also suggested a likely association and a high percentage of phenotypic variability between BW resistance and PH (Figure 2B). The dimension of the first PC (Dim1) largely delineated and explained 63.8% of the phenotypic variability for BW resistance and PH (Figure 2B). The dimension of the second PC (Dim2) illustrated the 36.2% phenotypic variability for BW resistance and PH at opposite angles of the PCA biplot (Figure 2B). This data also showed that BW resistance and PH are controlled by multiple genes and have a negative correlation.

### 3.4. Linkage Map Construction of the F_2_ Population

A total of 1485 SNP markers were polymorphic between the parents. Among them, 378 were used based on the segregation ratio and missing data (Table 1). Those 378 markers were used to construct the linkage map. A linkage map was constructed using these markers, which spanned a genetic distance of 1001.3 cM (Table 1). The map results produced a total of 12 linkage groups, which are equivalent to the number of tomato chromosomes. The individual chromosomes encompass 16 to 77 markers with lengths ranging from 70.7 to 128.1 cM across the 12 chromosomes (Table 1). The highest average distance between the markers was recorded at 6.3 cM on chromosome 7 (Table 1). The lowest average distance between the markers was recorded at 1.1 cM on chromosome 11 (Table 1). The heat map shows steady heat across the transverse line within the chromosomes, indicating that the linkage map was constructed precisely (Figure 3).

### 3.5. QTL Mapping for Bacterial Wilt Resistance and Plant Height

Multiple QTLs were identified for BW resistance and PH variation using 140 F2 individuals and the SNP-based linkage map. For BW resistance, a total of three QTLs were detected on chromosomes 1, 2, and 3 (Figure 4 and Table 2). Two minor QTLs, qbw-01 and qbw-02, on chromosomes 1 and 2 had LOD scores of 2.7 and 2.5 and explained 8.8% and 3.6% of the phenotypic variation for the BW resistance (Figure 4 and Table 2). One major QTL qbw-03 on chromosome 3 had a LOD score of 3.1 and explained 14.9% phenotypic variation (Figure 4 and Table 2). For the PH variation, a total of three QTLs were detected on chromosomes 1, 8, and 11 (Figure 4 and Table 2). One minor QTL qph-01 on chromosome 1 had a LOD score of 3.2 and explained 7.1% of the phenotypic variation. Two major QTLs, phq-08 and phq-11, on chromosomes 8 and 11, had LOD scores of 4.5 and 4.8 and explained 11% and 10.4% of the phenotypic variation for the PH variation (Figure 4 and Table 2).

For further validation of the QTLs, the markers linked to the BW-resistant QTLs were used to draw the box plots to visualize the QTLs’ contribution to resistance. For the QTLs qbw-02 and qbw-03, the level of BW resistance of the F_2_ plants that harbored the homozygous allele from CLN1466EA was significantly higher than that for plants carrying the heterozygous and homozygous alleles of the NC 84173 parents (Figure 5). Likewise, for the PH variation, F_2_ plants that harbored the homozygous allele from NC 84173 were taller than those plants carrying the heterozygous and homozygous alleles of the parents, CLN1466EA (Figure 5). These results indicated that those QTLs and linked markers could be useful resources for the breeding of BW resistance and PH-related traits.

## 4. Discussion

A segregated population was developed and evaluated for BW resistance and PH under greenhouse conditions. Phenotype evaluation revealed that both traits are controlled by multi-genes. Microarray Beadchip-derived SNPs were used to construct linkage maps and perform QTL analyses, which identified three QTLs for BW resistance and three for PH variation. Previous studies have shown that BW resistance in tomatoes is controlled by multiple genes in the field and greenhouse conditions [26,34,37,42]. Our resistance evaluation results were also following the previous reports showing the continuous distribution in a segregating population.

BW resistance in tomatoes can be specific to the location and specific to a strain; also, it fluctuates depending on screening methods [24,63,64]. Temperature-mediated resistance has also been reported [65]. ‘HI 7996’ (*Solanum lycopersicum*) is one of the most durable sources of resistance against BW to use in breeding new tomato varieties. This line showed a high survival rate (97%) over 12 field assays carried out in 11 countries in America, Asia, Australia, and the Indian Ocean region [29]. The present study was conducted in a protected environment, and the parental line CLN1466EA was used as a source of resistance. A previous report showed that line CLN1466EA had an introgression sequence of ‘HI 7996’ (*Solanum lycopersicum*) on chromosome 12 [66]. However, in our study, we did not detect any resistance factor on chromosome 12. This might be due to the strain we used in the present study or a different set of markers.

Several marker types have been used to map BW resistance in tomatoes, including simple sequence repeat (SSR), sequence-characterized amplified region (SCAR), amplified fragment length polymorphic (AFLP), cleaved amplified polymorphic sequence (CAPS), and restriction fragment length polymorphism (RFLP) markers [26,34,36,37,38,39,40,41,42,43]. Recently, single nucleotide polymorphism (SNP) markers have become an abundantly used marker system in mapping studies [67,68,69]. Out of 12,654 genotyping-by-sequencing-based SNPs, 1404 were used to construct a genetic linkage map that spanned 1322 cM with a 0.94 cM average genetic distance between two adjacent markers [67]. In the present study, we used the microarray Beadchip-based SNPs to construct a linkage map. Due to low polymorphism and missing data, only 378 out of 1485 SNPs were used to construct the map. However, the linkage map spanned 1001.2 cM, which is comparable with previous reports [44,67,70].

Numerous studies have been conducted to map resistant QTLs against BW in tomatoes [26,34,37,39,42,44,67,71,72]. Inoculation technique-specific QTLs were detected on chromosomes 6 and 10 (R^2^ values of 77.3% and 24.6%, respectively) in plants assayed using roots, and on chromosomes 6, 7, and 10 (R^2^ values of 30.2%, 24.4%, and 38.2%, respectively) when assayed using shoots [38,72]. Four QTLs, namely, Bwr-3, Bwr-4, Bwr-6, and Bwr-8, were detected using an F_2:3_ and RIL population, explaining phenotypic variation spanning between 3.2 and 29.8%. Among these, the major QTL Bwr-6 and minor QTL Bwr-3 were consistently identified across the different environments [37,42]. The BW-resistant QTL on tomato chromosome 3 was also confirmed by a whole genome sequencing-based study in which a marker Bwr3.2dCAPS positioned in the Asc (*Solyc03g114600.4.1*) gene showed substantial connotation with BW resistance [73]. In our study, we identified a major QTL qbw-03 on chromosome 3, contributing to the phenotypic variation of 14.9% with a LOD score of 3.1. In another study, two major QTLs, *Bwr-12* (R^2^ value of 17.9% to 56.1%) and *Bwr-6* (R^2^ value of 11.5% to 22.2%), on chromosomes 12 and 6 were detected using 188 RILs [42]. However, in our study, we did not identify any QTLs on chromosomes 6 and 12. This might be due to the use of different strains, phenotyping methods, or sets of markers used in the present study.

A recent study used an image-based non-destructive phenotyping method to map QTLs against BW [44]. They identified QTLs associated with BW resistance on chromosomes 2, 3, 4, 6, 8, 10, and 12, with R^2^ ranging between 5.38 and 11.46% [44]. In our study, we identified QTLs on chromosomes 1, 2, and 3, which are close to the reported QTLs in the aforementioned study. However, we cannot compare the QTLs precisely due to a lack of information about the physical positions of the linked markers on the genetic maps. Furthermore, the previous researchers used different plant populations with different sources of resistance and also used different isolates of *Ralstonia solanacearum*.

QTL mapping for PH was also performed in the present study. A study reported one locus associated with tomato PH on chromosome 2 [49]. Another study detected several loci associated with tomato PH on chromosomes 3, 4, 9, 11, and 12 through an introgression line population developed from the cross between *S. chmielewskii* and *S. lycopersicum* [51]. In our study, we also identified a major QTL on chromosome 11 (*qph-11*) contributing to the PH (R^2^ value of 10.4%). In a different QTL mapping study, three loci for PH were identified, in which *h4t2a* and *h4t3a* reduced PH, whereas *h4t7a* increased PH [52]. Furthermore, a recent study reported seven QTLs on chromosomes 1, 3, 5, 9, 10, 11, and 12, named *qtph1.1*, *qtph3.1*, *qtph5.1*, *qtph9.1*, *qtph10.1*, *qtph11.1*, and *qtph12.1* [45]. The major QTL in that study was detected on chromosome 1, which was further fine-mapped for gene identification [45]. In our study, we also identified QTLs linked to tomato PH on chromosomes 1 (*qph-01*) and 11 (*qph-11*). Another recent study that used an image-based non-destructive phenotyping method reported QTLs controlling plant architecture on chromosome 8 named *Tpa8.1* [44]. In the present study, we detected a major QTL on chromosome 8 (*qph-08*) associated with PH (R^2^ value of 11%). However, due to the different phenotyping methods and different sets of markers, we are unable to compare the precise locations of the QTLs. Further fine-mapping studies are required to narrow down these QTL regions using advanced inbred lines in multi-locations.

## 5. Conclusions

In conclusion, a segregating population was developed using two contrasting parental lines, CLN1466EA (BW-resistant and short in height) and NC 84173 (BW susceptible and tall in height), to map the loci associated with BW resistance and PH in tomatoes. We used microarray Beadchip-derived SNPs to construct a genetic map and identified a total of six loci, three for BW resistance and three linked to PH. These results can be useful in developing molecular markers linked to BW resistance and PH (architecture-related traits) for marker-assisted selection and to accelerate tomato breeding programs.

## Figures and Tables

**Figure 1 plants-13-00876-f001:**
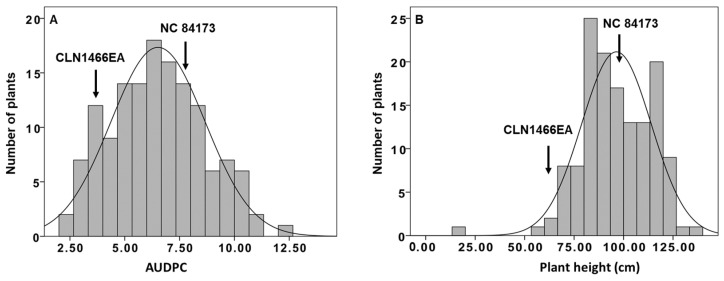
Frequency histograms of the collected phenotypic dataset of the developed mapping population. (**A**) The frequency distribution of the area under the disease progress curve. (**B**) The frequency distribution of the plant height.

**Figure 2 plants-13-00876-f002:**
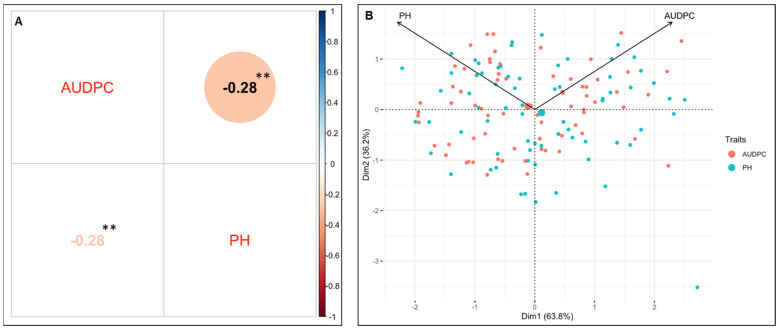
Phenotypic analysis of the developed segregated mapping population. (**A**) Pearson’s correlation analysis, and (**B**) principal component analysis (PCA). In the figure, the symbol ‘**’ indicates significance at *p* < 0.01.

**Figure 3 plants-13-00876-f003:**
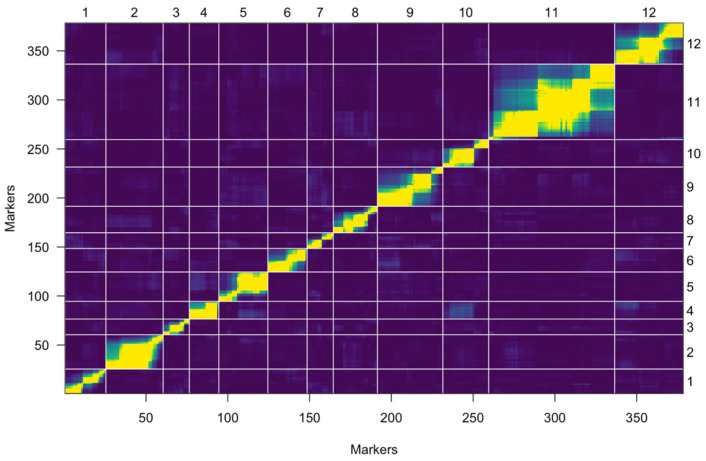
Heat map of a constructed genetic linkage map in a developed mapping population, NC 11212. The X-axis denotes the arranged markers and chromosome numbers. The Y-axis displays the recombination frequency and LOD scores for all pairs of markers.

**Figure 4 plants-13-00876-f004:**
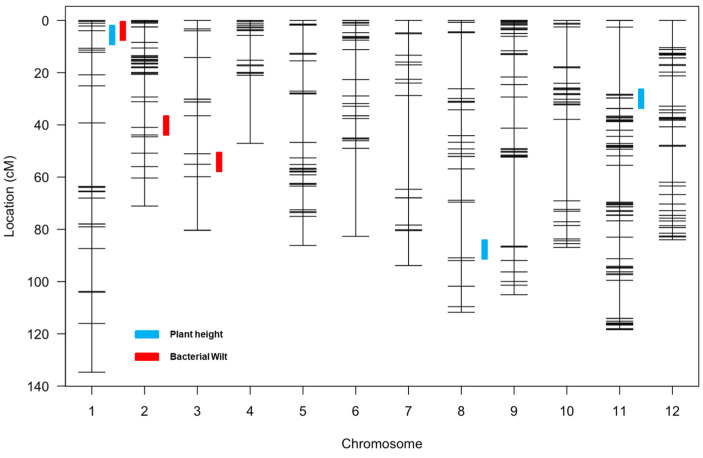
Genetic linkage map based on SNP genotyping in mapping population NC 11212 and detected QTLs for bacterial wilt resistance and plant height.

**Figure 5 plants-13-00876-f005:**
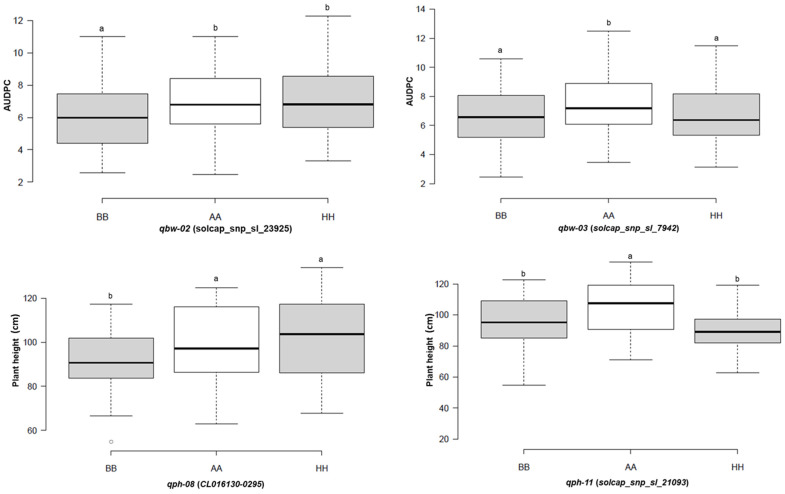
Box plots of closely linked markers associated with bacterial wilt resistance and plant height. Small letters (a and b) refer to significant differences (*p* < 0.05) according to Duncan’s multiple range test. BB, the genotype of the paternal parent; AA, the genotype of the maternal parent; HH, the genotype of the heterozygote.

**Table 1 plants-13-00876-t001:** Summary of the linkage maps of NC 11212 showing 12 chromosomes along with the number of markers per chromosome and the length of each chromosome.

Chromosomes	Markers	Genetic Distance (cM)	Average Marker Interval (cM)	Physical Distance (Mb)
Chr1	25	128.1	5.3	107.2
Chr2	35	72.6	2.1	84.8
Chr3	16	79.9	5.3	83.6
Chr4	18	48.6	2.8	77.4
Chr5	30	70.7	2.4	72.4
Chr6	24	78.2	3.4	56.1
Chr7	16	94.7	6.3	74.8
Chr8	27	92.9	3.5	62.5
Chr9	40	87.4	2.2	75.4
Chr10	28	88.3	3.2	77.6
Chr11	77	88.6	1.1	60.2
Chr12	42	71.3	1.7	68
	378	1001.3	3.275	900

**Table 2 plants-13-00876-t002:** QTL associated with bacterial wilt disease resistance in an intraspecific mapping population, NC 11212, of tomatoes.

Trait	QTLs	Chromosomes	Position (cM)	LOD	R^2^	Additive
BW	*qbw-01*	1	2.2	2.7	8.8	0.70
BW	*qbw-02*	2	42.9	2.5	3.6	−0.38
BW	*qbw-03*	3	54.2	3.1	14.9	0.09
PH	*qph-01*	1	4.1	3.2	7.1	−0.20
PH	*qph-08*	8	90.7	4.5	11.0	0.27
PH	*qph-11*	11	32.8	4.8	10.4	−0.20

## Data Availability

All data are available within the manuscript.

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
