# Peer review of "Quantitative Trait Loci Mapping for Bacterial Wilt Resistance and Plant Height in Tomatoes"

_plants, 2024, doi:10.3390/plants13060876_

Round 1

Reviewer 1 Report

Comments and Suggestions for Authors

This manuscript (MS) performed QTL analysis for bacterial wilt resistance (BW) and plant height (PH) in tomatoes. There were 6 QTLs with different effects were identified with 3 QTLs for each trait. The results of this MS are useful for marker-assisted selection breeding in tomatoes targeted on BW and PH. Following are some minor suggestions.

1. It would be great if the author could add the morphological pictures of two parents showing their differences in BW and PH.

2. For Figure 1B and Figure 5, plant height should have its unit ‘cm’.

3. Typos (commas) in lines 310 to 311.

4. Gene and Latin names should be written in italics.

5. Please correctly use the units ‘μLmL’ rather than ‘μlml’. Consistently using ‘intraspecific’ or ‘intra-specific’

6. ‘5. Conclusions ’

Author Response

Thank you for the valuable comments to improve the manuscript. Please refer to the attached file below for the answers to the comments. Thanks. 

Reviewer 2 Report

Comments and Suggestions for Authors

I truly enjoyed reading your paper--found it to flow well, right from the abstract to your conclusions. I did make a few comments in the text (attached)--mostly about how I appreciated the way (in your results) that you drew inferences based on each tranche of data!  Several comments are included in the attached mark-up of your manuscript. I have a pair of modest concerns:  i) I am not sure that I understand, after reading all the way through your paper, why PH is an appropriate "pairing" for your BW SNP work. Perhaps that simply reveals my lack of understanding of how tomato breeding (and tomato production?) functions? Or perhaps, appropriate plant architecture, which might be your intended target, is too complex to address even in a QTL context? And so, PH becomes a reasonable proxy for plant architecture?  As you can see both here and within the attached mark-up, I struggled with this question. My other concern was perhaps also a misunderstanding on my part. You found QTLs which accounted for up to 10 or 11% of the total variance in both BW and PH. Is that "good" compared with other such studies? Based on what I read in your discussion, perhaps other work has found higher levels of variance accounted for in the realm of BW. But maybe people haven't done this so much with PH? Seems that your PH paragraph was focused on which chromosome might be involved--and perhaps that more fundamental concern makes a given QTL less of a focal point?

Author Response

(The authors gave the same response as above.)
